# Functional and Conformational Plasticity of an Animal Group 1 LEA Protein

**DOI:** 10.3390/biom12030425

**Published:** 2022-03-10

**Authors:** Brett Janis, Clinton Belott, Tyler Brockman, Michael A. Menze

**Affiliations:** Department of Biology, University of Louisville, Louisville, KY 40292, USA; brett.janis@louisville.edu (B.J.); cjbelott@gmail.com (C.B.); tyler.brockman@louisville.edu (T.B.)

**Keywords:** protein condensate, water stress, cryptobiosis, extremophiles, late embryogenesis abundant, LLPS

## Abstract

Group 1 (Dur-19, PF00477, LEA_5) Late Embryogenesis Abundant (LEA) proteins are present in organisms from all three domains of life, Archaea, Bacteria, and Eukarya. Surprisingly, *Artemia* is the only genus known to include animals that express group 1 LEA proteins in their desiccation-tolerant life-history stages. Bioinformatics analysis of circular dichroism data indicates that the group 1 LEA protein *Af*LEA1 is surprisingly ordered in the hydrated state and undergoes during desiccation one of the most pronounced disorder-to-order transitions described for LEA proteins from *A. franciscana*. The secondary structure in the hydrated state is dominated by random coils (42%) and β-sheets (35%) but converts to predominately α-helices (85%) when desiccated. Interestingly, *Af*LEA1 interacts with other proteins and nucleic acids, and RNA promotes liquid–liquid phase separation (LLPS) of the protein from the solvent during dehydration in vitro. Furthermore, *Af*LEA1 protects the enzyme lactate dehydrogenase (LDH) during desiccation but does not aid in restoring LDH activity after desiccation-induced inactivation. Ectopically expressed in *D. melanogaster* Kc167 cells, *Af*LEA1 localizes predominantly to the cytosol and increases the cytosolic viscosity during desiccation compared to untransfected control cells. Furthermore, the protein formed small biomolecular condensates in the cytoplasm of about 38% of Kc167 cells. These findings provide additional evidence for the hypothesis that the formation of biomolecular condensates to promote water stress tolerance during anhydrobiosis may be a shared feature across several groups of LEA proteins that display LLPS behaviors.

## 1. Introduction

Since their discovery in cotton seeds by Dure et al. [1], the importance of late embryogenesis abundant (LEA) proteins in the abiotic stress tolerance of plants has been firmly established. However, their actual function(s) are still largely unresolved [2,3,4]. Research over the past two decades has extended the role(s) of these intrinsically disordered proteins (IDPs) in promoting desiccation tolerance to animals [5,6]. Based on conserved sequence motifs, several LEA classifications schemes have been proposed, resulting in 6 to 11 distinct protein groups or families [7]. Using the classification proposed by Tunnacliffe and Wise, most investigated anhydrobiotic animals express LEA proteins from Group 3 (PF02987) [3]. The only exception is the brine shrimp, *Artemia franciscana*, which expresses LEA proteins from two additional families. These LEA proteins are from Group 1 (PF00477) and Group 6 (PF04927) and are also only expressed in the anhydrobiotic encysted embryos [5,8,9,10]. Compared to other anhydrobiotic animals, the reasons for the more extensive LEA repertoire in *Artemia* are still unresolved. However, despite a uniquely broad range of expressed LEA proteins, the knockdown of Group 1 LEA proteins by RNAi reduced the cyst desiccation tolerance by over 90% [11]. Therefore, expressed proteins from other groups do not readily substitute the anhydrobiotic functions performed by Group 1 LEA proteins in *A. franciscana*. Understanding the mechanism(s) by which *Af*LEA1 improves water-stress tolerance in *A. franciscana* may facilitate the development of methods to engineer anhydrobiotic research cell lines and water-stress tolerant medical stem cells.

A minimum of five similar Group 1 LEA proteins are concurrently expressed in *A. franciscana* and localize to the cytosol and the mitochondrion [10,11,12,13]. These proteins contain 2 to 8 repeats of a Group 1 amino acid motif (GGOTRREQLGEEGYSQMGRK) and, based on the EST data available at the National Center for Biotechnology Information (NCBI), are composed of 62 to 217 amino acids [10]. The observation of Group 1 LEA proteins being present in both the cytosol and the mitochondria may offer insight into their role(s) in desiccation tolerance. The occurrence of nearly identical protein variants in several cellular locations makes it unlikely that these proteins are highly selective in the targets they protect and may indicate a more general mechanism of protection conferred to multiple compartments and targets [10,11]. Furthermore, the reason(s) for similar proteins of varying sizes have not been explained. The observed variations in length may indicate that the number of repeating Group 1 motifs is not critical to the protein’s overall function. Alternatively, the number of repeats may regulate the behavior of a specific Group 1 protein during desiccation. Increased numbers of Group 1 motifs in each polypeptide could increase the odds of protein folding during water removal or increase the number of binding sites for desiccation-sensitive targets. On the other hand, shorter proteins should be more mobile in the drying cell’s increasingly crowded milieu than large polymers. The presence of protective proteins with different mobilities might be advantageous for reaching desiccation-sensitive targets at varying hydration levels and cytoplasmic viscosity [14,15,16].

Despite over a decade of study focused on Group 1 LEA proteins in *Artemia*, little is known about their functional mechanism(s) of protection. The sequence features of *Af*LEA1 have been investigated using bioinformatics, and in vivo experiments have been performed by transfecting *E. coli* and *S. cerevisiae* cells with *Af*LEA1 variants [13,17]. In these experiments, no improvements in osmotic stress tolerance could be detected for *E. coli,* and modest improvements in viability after freezing and drying were observed in yeast [13]. Similar observations were made after expressing a mitochondrial-targeted *Af*LEA1 variant from *A. franciscana* in Kc167 cells from *D. melanogaster*. In this system, the protein led to modest improvements in osmotic stress tolerance and conferred some protection to mitochondrial functions during freezing [12]. As with other LEA proteins, determining the mechanism by which Group 1 LEA proteins confer desiccation tolerance and identifying their desiccation-sensitive targets is more straightforward using in vitro techniques than transgenic animals [18]. However, the unique challenge of performing biochemistry at low water contents or in the desiccated state has led to limited progress on all fronts.

Nonetheless, methods have been developed to deduce the secondary structure of LEA proteins in the desiccated state using bioinformatics and circular dichroism [19,20]. Recently, the liquid–liquid phase separation (LLPS) behavior of a Group 6 LEA protein was investigated using a combination of light microscopy, scanning electron microscopy, atomic force microscopy, and drying procedures in the presence of osmolytes found in the anhydrobiotic cyst of *A. franciscan*a [21]. Utilizing similar techniques and guided by computation tools, we performed a comprehensive analysis of the behavior of *Af*LEA1 during desiccation, and our findings provide new insights into two specific hypotheses regarding LEA protein functions: molecular shielding [22] and hydration buffering [23].

## 2. Material and Methods

### 2.1. Chemicals

All chemicals, including those for transgenic protein expression, preparation of buffer solutions, and measuring lactate dehydrogenase activity, were purchased from Sigma-Aldrich (St. Louis, MO, USA) or obtained from VWR (Atlanta, GA, USA). Water for solution preparation was purified with a custom-built reverse osmosis water system (Culligan Water, Clarksville, IN, USA). Water for molecular biology applications was purchased from VWR (Atlanta, GA, USA).

### 2.2. Protein Cloning, Expression, and Purification

DNA encoding for the 180 amino acids-long *Af*LEA1 (highly similar to ABR67402; described in: [12]) polypeptide was cloned into the Ptxb1 (NEB Biolabs, Ipswich, MA, USA) vector and expressed using isopropyl-β-1-thiogalactopyranoside (IPTG) induction as previously described [24]. Briefly, BL21 Star strain *Escherichia coli* cells transformed with this vector were cultured on agarose plates containing 100 µg/mL ampicillin in Luria Bertani (LB) medium. For protein induction, single colonies were grown to an optical density of ~0.6 at λ = 595 nm overnight at 37 °C and cells were harvested 2 h after adding 0.4 mM IPTG. The bacteria were harvested at 5000× *g* for 30 min at 4 °C and the pellets were resuspended in buffer A (500 mM NaCl, 50 mM Tris-HCl, pH 8.5) containing 1 mM phenylmethylsulphonyl fluoride (PMSF) and stored at −80°C until protein purification.

For protein purification, the bacterial pellets were thawed and sonicated (Q500, Qsonica, Newtown, CT, USA), then centrifuged at 20,000× *g* for 30 min at 4 °C to remove cellular debris. The resulting supernatant was applied to a column containing 15 mL chitin resin (NEB Biolabs, Ipswich, MA, USA) and the column was washed with 3 column volumes of buffer A. Self-cleavage of the intein protein was induced by adding 50 mM dithiothreitol at 4 °C for 48 h and the eluted protein was dialyzed overnight against 20 mM TRIS-HCl at a pH of 8.0. Further purification was performed using a fast-performance liquid chromatography (FPLC) system (AKTA, Cytiva Life Sciences, Marlborough, MA, USA) and a 1 mL Resource Q anion exchange column (Cytiva Life Sciences, Marlborough, MA, USA). Bound *Af*LEA1 was eluted from the column using a gradient of 20 mM TRIS-HCl at pH 8.0 and NaCl concentrations from 0 to 125 mM over 10 column volumes. *Af*LEA1 eluted from the column was 99% pure based on band intensity measured using SDS PAGE and the protein was dialyzed against 50 mM phosphate buffer, pH 7.0. Aliquots of purified *Af*LEA1 were snap-frozen in liquid N_2_ and stored at −80 °C until used for experimentation.

### 2.3. Size-Exclusion Chromatography

Size-exclusion chromatography was performed using a Superdex 75^®^ 10/300 column on an FPLC instrument (AKTA, Cytiva Life Sciences, Marlborough, MA, USA). A volume of 100 µL *Af*LEA1 protein in 50 mM phosphate buffer at pH 7.0 was injected at a 0.1 mL/min flow rate. Low molecular weight standards (Cytiva Life Sciences, Marlborough, MA, USA), including aprotinin (6.5 kDa), ribonuclease A (13.7 kDa), carbonic anhydrase (29 kDa), ovalbumin (44 kDa), conalbumin (75 kDa), and a blue dextran 2000 tracking polymer, were used as molecular weight markers.

### 2.4. Circular Dichroism

*Af*LEA1 protein (100 µg/mL in 20 mM phosphate buffer at a pH of 7.0) was measured in a sealed quartz cuvette with a path length 0.1 cm (Starna Scientific, Atascadero, CA, USA) using a wavelength range from 280 nm to 185 nm using a J-1500 circular dichroism spectrophotometer (Jasco, Easton, MD, USA). To reduce the light-scattering associated with a dried sample, *Af*LEA1 was dialyzed into ultrapure water three times at a ratio of 1:1000 protein solution to water. Next, *Af*LEA1 at 1 mg/mL in water was repeatedly plated on an open 0.01 cm path-length quartz cuvette (Starna Scientific, Atascadero, CA, USA). Each layer was rapidly dried by incubating them for 1 h at 0% relative humidity over anhydrous calcium sulfate. The rapid drying process produced an amorphous protein glass with minimal light scattering. Once the protein layer reached the thickness of the 0.01 cm path length, 1 µL of ultrapure water was added, allowing the cuvette to be assembled. A final incubation for 24 h at 25 °C and 0% relative humidity produced a protein glass of *Af*LEA1 without observable light scattering. Data were averaged over 5 measurements taken in 1 nm intervals. Secondary structure predictions were performed using the CONTIN and SELCON 3 predictors from DichroWeb (http://dichroweb.cryst.bbk.ac.uk/html/home.shtml, last accessed on 20 February 2022) with data sets 4 and 7 as references.

### 2.5. Bioinformatics Structural Predictions

I-Tasser (https://zhanggroup.org/I-TASSER/, last accessed on 20 February 2022) predicts the structure of proteins by comparing the portions of known crystal structures available from the protein data bank (PDB, https://www.rcsb.org, last accessed on 20 February 2022) to the queried polypeptide and combining them into models using a hierarchical ranking system [25,26,27]. The protein modeling software Swiss PDBViewer (DeepView) (https://spdbv.unil.ch, last accessed on 20 February 2022) was used to visualize the potential hydrophobic face of the *Af*LEA1’s α-helices [28]. The potential for phase separation of *Af*LEA1 was evaluated using catGranule (http://service.tartaglialab.com, last accessed on 20 February 2022) [29].

### 2.6. Sample Preparation for Light Microscopy and Electron Scanning Microscopy (SEM)

A droplet of *Af*LEA1 in H_2_O was plated onto microscope slides directly adjacent to a droplet of ultrapure water. The droplets were then connected to allow diffusion of the protein and produce a protein gradient over 10 min at ambient relative humidity and 25 °C. The samples were dried at 0% relative humidity over anhydrous calcium sulfate at 25 °C to remove any residual water. Glass coverslips were placed above the samples and sealed with nail polish after desiccation-preserving the samples. The samples were stored in a sealed container at 0% RH until viewed using a specimen microscope. Sample preparation for SEM was performed as described above, with the difference that an aluminum SEM stage was used to deposit the protein solution. Once completely dried, the samples were sputter-coated with a 10 nm layer of gold to prevent sample rehydration reduce charging artifacts. The samples were stored at 0% RH until viewed using an electron microscope (Jeol, Peabody, MA, USA).

### 2.7. Lactate Dehydrogenase Activity Assays 

Cell lysates were obtained by sonication of 20 × 10^6^ Kc167 cells (*Drosophila melanogaster*) in 100 mM phosphate buffer at a pH of 6.4. Cell debris was removed by centrifugation at 15,000× *g* for 10 min. The supernatant was diluted to a total protein concentration of 2 mg/mL with phosphate buffer, and the total protein concentrations were confirmed via Bradford assays. *Af*LEA1 or BSA was then added to the lysate to yield a total concentration of 400 μg/mL of *Af*LEA1 or BSA, and the endogenous LDH activity was recorded. Aliquots of 50 μL were then desiccated for 7 days in a sealed container at 25 °C and 0% RH. Samples were rehydrated with 100 μL phosphate buffer, and activity was measured using UV-VIS spectrophotometry (UV-1800, Shimadzu, Tokyo, Japan). Absorbance was monitored at λ = 340 nm using the kinetics mode while the sample was stirred at 500 rpm, and the sample temperature was maintained at 25 °C.

To distinguish between protection of LDH and repair of LDH by *Af*LEA1, pure LDH was desiccated in the presence or absence of 400 μg/mL *Af*LEA1 or BSA in 100 mM sodium phosphate buffer at a pH of 6.4. Commercially obtained LDH (Millipore-Sigma, St. Louis, MO, USA) was diluted to 0.2 mg/mL with H_2_O and dialyzed against 100 mM phosphate buffer, pH 6.5. Initial LDH activity was determined, and 25 μL aliquots of samples containing LDH, LDH plus BSA, or LDH plus *Af*LEA1 were placed in microtubes and desiccated for 7 days at 0% RH. Samples were rehydrated with 50 μL of phosphate buffer. Additional samples of purified LDH were desiccated in the absence of *Af*LEA1 and BSA for 7 days at 0% RH and rehydrated on ice with either 50 μL of phosphate buffer or phosphate buffer containing *Af*LEA1 or BSA at 400 μg/mL to investigate the potential repair mechanisms. LDH activity after rehydration was measured as described above.

### 2.8. Screening AfLEA1 for RNA-Induced Liquid-Liquid Phase Separation

A solution resembling the crowded cytoplasm of the diapause cysts of *A. franciscana* (32 mM NaCl, 98 mM KCl, 11 mM K_2_PO_4_, 5 mM CaCl_2_, 340 mM trehalose, 2.9% *w*/*v* glycerol, and 25% Ficoll 400, pH of 6.5) was prepared, and 150 μg/mL of *Af*LEA1 was added to determine whether *Af*LEA1 undergoes LLPS during desiccation. The protein solution was deposited onto microscope slides, and the solutions were allowed to desiccate at ambient humidity (83% RH) and temperature (22–26 °C) while observed using an inverted microscope. RNA (20 ng/mL final concentration) isolated from *A. franciscana* using a total RNA isolation kit (Qiagen, Germantown, MD, USA) was added to the protein droplet to determine the impact on the LLPS behavior of *Af*LEA1. 

### 2.9. SDS-PAGE

Samples of *Af*LEA1 obtained after affinity or ion-exchange chromatography were dialyzed against 50 mM phosphate buffer at a pH of 7.0 and frozen in 10 mL aliquots. Before electrophoresis, the protein was boiled in Laemmli buffer (Bio-Rad, Hercules, CA, USA), and 2 μg of *Af*LEA1 was run per lane of a 10% SDS-PAGE (37.5:1, Bio-Rad, Hercules, CA, USA). A kaleidoscope protein ladder was used as a molecular weight standard (Bio-Rad, Hercules, CA, USA). The gel was run at 90 V for 1 h using a running buffer containing 25 mM Tris, 192 mM glycine, and 0.1% SDS at a pH of 8.3, and gels were destained overnight with 30% methanol and 10% glacial acetic acid in H_2_O.

### 2.10. Cell Culture, Transgenic Cell Line, and Confocal Microscopy

Cell culture, transgenic cell line development, and confocal microscopy were performed as previously described for *Afr*LEA6 with the following differences [21]. In place of Schneider’s media, IPL-41 insect culture media (Gibco, Waltham, MA, USA) was supplemented with 2 g/L tryptone phosphate broth (VWR, Atlanta, GA, USA) and 10% heat-inactivated FBS (Atlanta Biologicals, Flowery Branch, GA, USA) and used to culture the Kc167 cells. In addition, transfected and control cell lines were stained for only 2 min with Nile Red to prevent fluorescence saturation in *Af*LEA1-expressing cells. Cells co-expressing *Af*LEA1-mCherry and GFP were stained with 200 µL of 100 nM MitoView Blue in DPBS (Biotium, Fremont, CA, USA) for 10 min. The following primers were used to clone and amplify untagged AfLEA1: 5′-CGGAATTCCAAACATGGAACTGTCGTCGAGTAAGCTG-3′ and 5′-TAATTGCGGCCGCATTTCTGTCTTGCGAGACCTCCTTTTTG-3′. The following primers were used to clone and amplify a chimeric protein composed of AfLEA1 and mCherry at the c-terminus: 5′-CGGCGAATTCATGGAACTGTCGTCGAGTAAGCTG-3′ and 5′-AGGGCGGCCGCATTTCTGTCTTGCGAGACCTCCTTTTTG-3′.

## 3. Results and Discussion

### 3.1. Protein Cloning, Expression, and Purification

The protein *Af*LEA1 was readily expressed in *E. coli,* and an approximately 90% purity could be reached using chitin-affinity chromatography, as shown by SDS-PAGE (Figure 1A). Surprisingly, when the crude protein sample was further purified using a strong anion exchanger, *Af*LEA1 elutes in two distinct fractions (Figure 1B). Still, the protein in both fractions migrates identically during SDS-PAGE and has a purity of over 99% (Figure 1A). The presence of two elution peaks containing *Af*LEA1 protein that behaves identically during electrophoresis may indicate higher-order assemblies that are lost during the denaturation step in electrophoresis or distinct conformational states undetectable by SDS-PAGE.

Bioinformatics suggests that *Af*LEA1 has high conformational plasticity [17], which supports the hypothesis that the protein may exist in more than one tertiary state when bound to the column’s polystyrene/divinylbenzene polymer matrix, causing two distinct elution fractions. Furthermore, a trend was observed indicating the concentration of salt required for elution of *Af*LEA1 correlates negatively with the quantity of *Af*LEA1 bound to the column for both elution fractions (Appendix A). This behavior may indicate interactions among *Af*LEA1 molecules on the column can change the affinity of the protein towards the strong anion exchanger for both conformational states. However, a more thorough investigation of this behavior would be needed to strengthen this interpretation.

The apparent molecular mass of *Af*LEA1 in both elution fractions was measured using size-exclusion chromatography to exclude SDS-sensitive higher-order oligomers (Figure 2). Independent of the fraction *Af*LEA1 eluted in during anion-exchange chromatography, size-exclusion chromatography indicates that *Af*LEA1 has an apparent molecular weight of 33 kDa, which is ~75% larger than the amino acid composition-based calculated mass (~19.7 kDa). However, intrinsically disordered proteins can have apparent molecular sizes up to 12 times as large as a similarly sized globular protein [30]. Overall, these results suggest *Af*LEA1 is likely in a molten globule state showing some secondary structure but not a tightly packed tertiary structure. In the molten globule state, when evaluated using size-exclusion chromatography, proteins can have an apparent molecular weight of up to twice that of a tightly packed globular protein of a similar length.

### 3.2. AfLEA1 Secondary Structural Analysis

The secondary structure of *Af*LEA1 was investigated by circular dichroism under three conditions to explore the conformational transition of *Af*LEA1 during the reduced availability of water (Figure 3). In the hydrated state, about 40% *Af*LEA1 contains defined secondary structure motifs, and the protein is composed of 5% α-helices, 35% β-sheets, 18% turns, and 42% random coils. While *Af*LEA1 can be classified, like other LEA proteins, as a mostly intrinsically disordered protein, it is surprisingly ordered in the hydrated state. For example, the Group 3 LEA proteins *Afr*LEA2 and *Afr*LEA3m (*A. franciscana*) are only ordered to 21% and 25%, respectively [31]. However, *Af*LEA1 undergoes a dramatic conformational transition in the desiccated state. The protein is nearly 100% ordered and composed of 85% α-helices, 5% β-sheets, and 10% turns after removing the water (Figure 3). Despite being more structured than most LEA proteins in the hydrated state, this conformational transition is the most dramatic shift from disorder to order reported for an LEA protein from *A. franciscana*. The transition from the hydrated to the desiccated conformation is partly mimicked by the molecular crowding effect observed in a solution containing 2% SDS (Figure 3). In the presence of SDS, *Af*LEA1 is slightly less ordered (38%) than in the hydrated state, and its β-sheets appear to transition into α-helices before the random coils. Generally, the secondary structure measurements and predictions produced by circular dichroism represent the average conformation of an ensemble of the different conformational states in the beam’s path [32]. It appears that during desiccation, *Af*LEA1 is transitioning into a more uniform conformation compared to the hydrated state, which is a strong indication of the disorder-to-order regulation of its protective functions.

The sequence of *Af*LEA1 is highly repetitive and is mainly composed of the eight repeats of the Group 1 consensus sequence ‘GGOTRREQLGEEGYSQMGRK’ [33,34] (Figure 4A). *Af*LEA1’s propensity for α-helices is predicted by the regular appearance of alanine-arginine-alanine “helix cap” motifs found at the ends of the repeating Group 1 LEA motif [35,36,37]. Given the amount of predicted α-helical structure, an α-helical protein projection is informative. Unlike *Afr*LEA2 and *Afr*LEA3m [38], *Af*LEA1 does not have sufficient secondary structure orientation of its charged residues to offer alpha-helical stabilization (Figure 4B) [39]. However, the protein does have a thin hydrophobic face, which can stabilize α-helical formation in the absence of water [40]. Nevertheless, the hydrophobic moment is too low to offer sufficient stabilization to explain *Af*LEA1’s α-helix content measured in the desiccated state [17,41]. 

### 3.3. Computational Interpretation of the Structure of AfLEA1 in the Desiccated State

The I-Tasser program uses X-ray crystallography data to predict protein structure. The program is surprisingly effective at predicting the ordered arrangements of LEA proteins that undergo conformational transitions during desiccation [25], and the secondary structure predictions of I-Tasser fall within 2% of the secondary structure content measured by circular dichroism for desiccated *Af*LEA1. The α-helices predicted by I-Tasser appear to be organized into a sizeable helical bundle composed of four helix-turn-helix structures, with each helix formed by a single Group 1 LEA consensus sequence and a short spacer containing a histidine residue (Figure 5A). The α-helices propagate from the alanine-arginine-alanine caps, as previously hypothesized [17]. Still, the internal space of the helical bundle is highly enriched in positively charged residues and aromatic residues (Figure 5B).

Although aromatic residues can interact with positively charged residues due to their delocalized π orbitals [42], the small number of aromatic residues does not appear to be sufficient to prevent the electrostatic repulsion of the α-helices in *Af*LEA1. Despite this source of instability, this structure is similar to a de novo-designed protein built to emulate armadillo repeat proteins (RCSB PDB ID: 5CWH), which has remarkably similar primary, secondary, and tertiary structures [43] to *Af*LEA1. Armadillo repeat proteins are highly stable and are commonly involved in cell signaling and misfolded protein degradation due to their promiscuous binding behaviors [44,45]. The structure of *Af*LEA1 appears to be stabilized by similarly charged stripes found on helices proposed for *Afr*LEA2 and *Afr*LEA3m [38]. The charge patterns are generated in these two LEA proteins from Group 3 through a specific arrangement of basic and acidic amino acids in the primary sequence (Figure 6A,B). If I-Tasser has accurately predicted the tertiary structure of *Af*LEA1, then charged stripes arise in this LEA protein from the tertiary association of alpha-helices, thereby stabilizing these helical clusters (Figure 6C) [17]. 

### 3.4. Behavior of AfLEA1 during Desiccation

Although *Af*LEA1 appears to evenly distribute itself as an amorphous, glassy deposition during fast desiccation in pure water, as used for circular dichroism, it undergoes crystallization when dried more slowly for hours rather than minutes (Figure 7A). The formation of crystals indicates that *Af*LEA1 assumes a uniform structure in the desiccated state, as indicated by the circular dichroism data and structural predictions by I-Tasser. Unfortunately, scanning electron microscopy (SEM) reveals that the crystals formed by *Af*LEA1 during slow drying are of insufficient quality to employ X-ray crystallography to verify the structure of *Af*LEA1 in the desiccated state (Figure 7B). However, SEM also revealed spherical structures in the desiccated sample similar to those reported for the Group 6 LEA protein *Afr*LEA6, which is known to undergo LLPS in vitro during desiccation [21].

The program catGranule predicts that *Af*LEA1 has an exceptionally high propensity to undergo LLPS, particularly when interacting with RNA (Figure 8A,B). Desiccation experiments were conducted in a buffer system that resembles the cellular environment of the encysted embryos of *A. franciscana* to determine whether *Af*LEA1 undergoes LLPS in a physiologically more relevant background. At high levels of desiccation, *Af*LEA1 does appear to undergo LLPS, but this occurs at nearly complete desiccation (Figure 9A), and at these extreme levels of water loss, the observed phenomenon may not be physiologically relevant. Behaviors such as salt-induced precipitation, which may cause proteins to separate from solution, are likely commonplace in the drying cell when protein–protein interactions become unavoidable [46]. The occurrence of LLPS at extreme dehydration may simple be a consequence of water loss, not a protective mechanism during anhydrobiosis. After adding total RNA isolated from *A. franciscana* to *Af*LEA1 samples before drying, the protein undergoes an LLPS readily after minor water removal by evaporative drying (Figure 9B). However, additional studies are required to verify that RNA is selectively incorporated into the *Af*LEA1 dense phase and if the *Af*LEA1 condensate offers protection against RNA degradation. 

The readily observed phase transition behavior of some LEA proteins may warrant revisiting the hydration buffering hypothesis. LLPS behavior involves the reordering of water as water molecules interact with the separating protein to produce a unique fluid with different surface tension, resulting in spherical protein-water droplets. The water contained in the protein-water phase can have properties that vary significantly from the bulk fraction of cytosolic water [47]. The hydration buffer hypothesis postulates that water molecules that are retained and interact at the surface of LEA proteins during dehydration serve as a water reservoir during desiccation [23,48]. In addition to LEA proteins serving as a water reservoir by organizing water molecules at the protein surface, a new organizational feature of water may occur inside of the protein-water droplet that protects desiccation-sensitive biomolecules at different stages of dehydration. Furthermore, changes in water-loss kinetics caused by some LEA proteins could be caused by the different physiochemical behavior of the water fraction in proteinaceous droplets. While associating biomolecular condensates with a general mechanism of desiccation tolerance still appears somewhat premature, research providing evidence for a significant role of proteinaceous phase transitions as part of the anhydrobiotic toolbox has begun to accumulate [6,17,21,49,50].

### 3.5. AfLEA1 Protection of Lactate Dehydrogenase (LDH) Activity during Desiccation and Rehydration

Although the above results may indicate that *Af*LEA1 is involved in RNA stabilization, one of the leading hypotheses for LEA protein function is the protection of other proteins and membranes during desiccation and rehydration via molecular shielding. Therefore, LDH was desiccated in the presence of *Af*LEA1 or BSA (Figure 10). Endogenous LDH activity in Kc167 cell lysates that lack LEA proteins was measured to simulate cellular conditions more closely than experiments using purified enzymes can achieve. A significantly higher residual LDH activity after rehydration was observed in samples containing *Af*LEA1, whereas BSA did not show any improvement over the control (Figure 10). This observation indicates that *Af*LEA1 offers protection against desiccation-induced protein denaturation, which could have been due to *Af*LEA1 having a chaperone-like refolding activity. Purified LDH was dried in the absence of *Af*LEA1 or BSA, and samples were rehydrated with a phosphate buffer or buffer containing either *Af*LEA1 or BSA to test if enzyme activity is higher if rehydrated with protein-containing solutions. Furthermore, the purified enzyme was dried in the presence or absence of these proteins and rehydrated with phosphate buffer. *Af*LEA1 preserved nearly 100% of LDH activity during desiccation if added before the onset of water removal but failed to affect LDH activity when only included in the rehydration solution (Figure 11). Therefore, *Af*LEA1 improved the desiccation tolerance through interactions with LDH during drying or in the desiccated state. BSA offered lesser protection against desiccation-induced loss of LDH activity to the purified enzyme than *Af*LEA1 when desiccated together with LDH and did not improve the LDH activity if only present during rehydration (Figure 11). It is noteworthy that the protection conferred by BSA to LDH activity was only observed for experiments using the purified enzyme but not in a more complex proteome sample. These results again demonstrate that the insights gained by experiments using purified targets are limited, and caution is warranted when extending these results to physiologically relevant contexts. 

### 3.6. AfLEA1 Expressed in Kc167 Cells from D. melanogaster Cells Localizes to the Cytoplasm and Improves Cellular Structural Integrity during Desiccation

Kc167 cells from *D. melanogaster* ectopically co-expressing *Af*LEA1-mCherry and GFP were stained with MitoView Blue and imaged using confocal microscopy. *Af*LEA1 was primarily found to localize to the cytoplasm and not the mitochondria, corroborating previously published results (Figure 12). Structures resembling proteinaceous condensates were observed in approximately 38% of *Af*LEA1-expressing cells when observed by confocal microscopy (Figure 12 and Appendix A). These structures were not detected in the previous work, utilizing conventional fluorescence microscopy [12]. The exclusion of GFP from these condensates suggests that this compartment may be an LLPS of *Af*LEA1, an interpretation strengthened by the observation that *Af*LEA1 in vitro readily undergoes an LLPS in the presence of RNA (Figure 9B). In addition, *Af*LEA1 appeared to accumulate in the nucleus in approximately 23% of cells (Figure 12 and Appendix A). We speculate that the formation of an *Af*LEA1 condensate and the protein accumulation in the nucleus may be due to interactions with nucleic acids that differ among cell cycle stages. However, additional cell-based studies exploring alternative expression and visualization approaches are needed to confirm any physiological significance of these observations.

During desiccation, Kc167 cells ectopically expressing untagged *Af*LEA1 displayed a robust increase in intracellular viscosity over the vector control cells when stained with Nile Red (Figure 13). Nile Red is a solvatochromatic dye that can increase in fluoresce intensity with increasing viscosity of the cytoplasm [51,52,53,54,55]. Increasing intracellular viscosity during modest dehydration likely helps impede molecular movement and prevent undesirable chemical reactions until more efficient glasses of protein and sugar can form [5,56,57]. Furthermore, the fusion of the plasma membranes during desiccation was significantly reduced in *Af*LEA1-expressing cells compared to the vector control. These results are like those obtained using *Afr*LEA6, possibly suggesting that both proteins utilize similar molecular mechanisms to confer protection during desiccation. However, despite sharing similar promiscuous protection mechanisms, different groups of biomolecules may be targeted by both proteins (e.g., proteins, nucleic acids, or lipids).

## 4. Conclusions

*Af*LEA1 and its protein variants in the animal extremophile *A. franciscana* are the only known Group 1 LEA proteins in this kingdom of life. Based on our findings, their highly repetitive structure aids in consistent protein folding behaviors during desiccation. The repetitive motifs in *Af*LEA1 may further act as multivalence sites for protein–protein and protein–RNA interactions. We speculate that this will be similar for LEA proteins from Group 1 found in other organisms. The readily observed phase separation of *Af*LEA1 in the presence of RNA in vitro strongly suggests a direct interaction between both biomolecules. However, *Af*LEA1 also improves the desiccation tolerance of proteins, such as LDH, making this protein promiscuous in its target selection. Based on these findings, we speculate Group 1 LEA proteins such as *Af*LEA1 may offer multiple modes of protection against desiccation-induced damage at different cellular hydration levels to various classes of biomolecules.

The reason(s) for multiple Group 1 LEA proteins that virtually only differ in their number of Group 1 repeats in *A. franciscana* is intriguing and requires further investigation. Similar Group 1 LEA proteins with different molecular mobilities may be advantageous during desiccation. Larger Group 1 LEA proteins may rapidly increase the viscosity of the cellular compartments at high water contents, thereby preventing protein aggregation and membrane collapse. In contrast, smaller Group 1 LEA proteins have higher mobility and can more readily interact with targets in this highly dense environment. We suspect the observed LLPS of *Af*LEA1 in the presence of RNA is a feature of other LEA proteins from Group 1, strengthening the notion that this recently observed phenomenon could be a common protective mechanism in the toolbox of anhydrobiotic organisms.

## Figures and Tables

**Figure 1 biomolecules-12-00425-f001:**
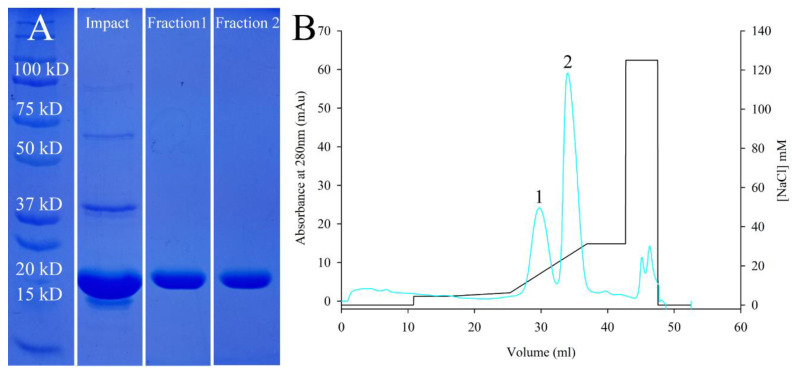
Purification of *Af*LEA1. (**A**) SDS-PAGE of crude samples obtained after affinity chromatography (IMPACT) or further purification using anion-exchange chromatography as a polishing step. Fractions 1 and 2 obtained during anion-exchange chromatography contain *Af*LEA1 at the expected molecular weight of ~19 kDa. (**B**) *Af*LEA1 binds to quaternary ammonium groups at a pH of 8.0 and elutes in two distinct fractions (blue line) at relatively low concentrations of NaCl (black line).

**Figure 2 biomolecules-12-00425-f002:**
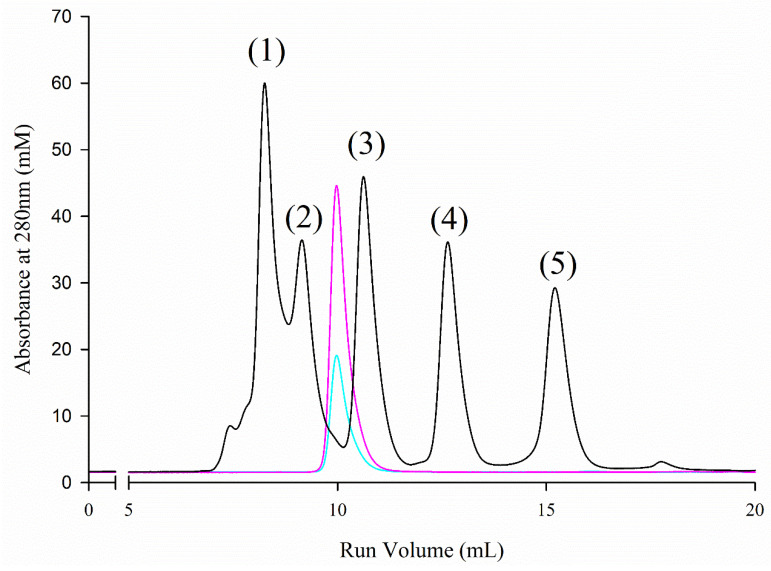
Size-exclusion chromatography of *Af*LEA1, both by anion-exchange chromatography, obtained fractions of the *Af*LEA1 (blue: peak fraction 1; magenta: peak fraction 2) elute from a size-exclusion column between ovalbumin (44 kDa) and carbonic anhydrase (29 kDa), respectively. Based on these data, the apparent molecular mass of *Af*LEA1 computes to 33.4 kDa, or approximately 75% larger than the calculated mass based on the polypeptide sequence of the protein. The black line shows the elution of molecular weight markers. The elution maxima correspond to: (1) conalbumin, 75 kDa; (2) ovalbumin, 44 kDa; (3) carbonic anhydrase, 29 kDa; (4) ribonuclease A, 13.7 kDa; and (5) aprotinin 6.5 kDa.

**Figure 3 biomolecules-12-00425-f003:**
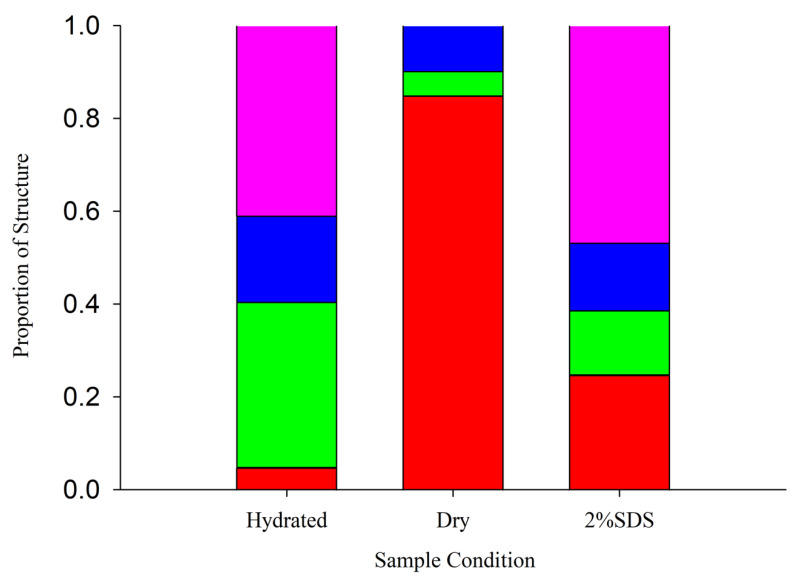
Circular dichroism analysis of hydrated and desiccated *Af*LEA1, wherein α-helices (red), β-sheets (green), turns (blue), and random coils (magenta) are represented as proportions of the total protein structure. In the hydrated state, the secondary structure of *Af*LEA1 was on average 5% α-helices, 35% β-sheets, 18% turns, and 42% random coils. In the desiccated state, the secondary structure of *Af*LEA1 was on average 85% α-helices, 5% β-sheets, and 10% turns. In the presence of 2% SDS, the secondary structure of *Af*LEA1 was on average of 25% α-helices, 13% β-sheets, 16% turns, and 46% random coils (for CD spectra see Appendix A).

**Figure 4 biomolecules-12-00425-f004:**
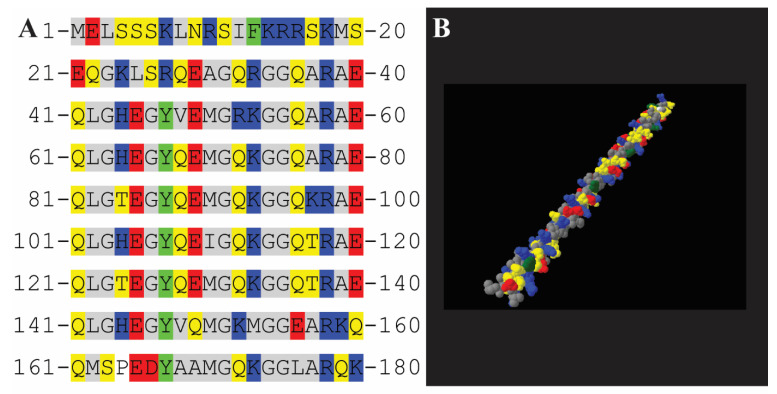
Amino acid sequence features of *Af*LEA1 where the amino acid properties are labeled by colors representing aromatic (green), polar (yellow), nonpolar (grey), positive (blue), and negative (red) amino acid side chains. (**A**) The amino acid sequence of *Af*LEA1 is highly repetitive and consists of eight Group 1 LEA domains. (**B**) Projected as an α-helix, *Af*LEA1 has a distinct hydrophobic face of nonpolar amino acids (gray) but does not have distinctly organized stripes of charged amino acid as described for some Group 3 LEA proteins.

**Figure 5 biomolecules-12-00425-f005:**
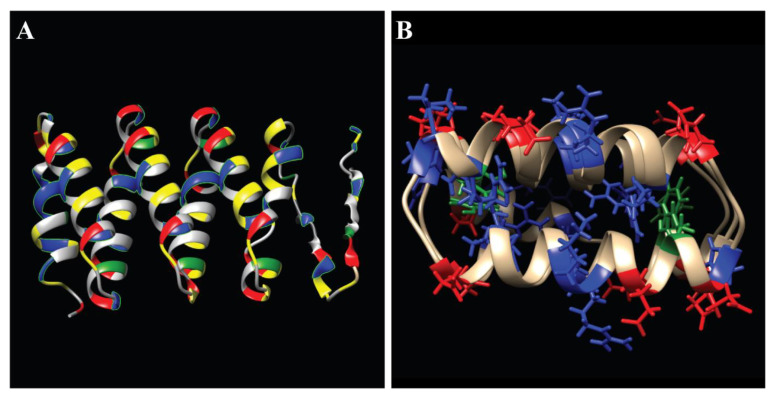
I-Tasser prediction of *Af*LEA1 structure in the desiccated state with amino acid properties labeled by color as polar (yellow), nonpolar (gray), aromatic (green), positive (red), and negative (blue). (**A**) *Af*LEA1 is predicted to fold into 84% α-helix, 5% β-sheet, and 11% turns. The tertiary structure of *Af*LEA1 is predicted to resemble a synthetic helical repeat protein (RCSB PDB ID: 5CWH) composed of helix-turn-helix structures, where each helix-turn-helix is a single Group 1 LEA motif. (**B**) The distribution of charged amino acids stabilizes the exterior of the *Af*LEA1. The protein’s interior is enriched with positive and aromatic residues.

**Figure 6 biomolecules-12-00425-f006:**
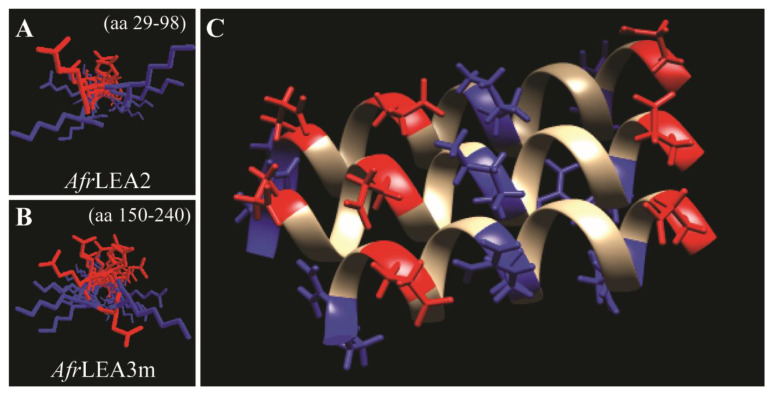
Three-dimensional structure and charge distribution of three LEA proteins from *A. franciscana* as predicted by I-Tasser. Colors represent the positive-charge residues in red and negative residues in blue. (**A**) Predicted α-helical regions of *Afr*LEA2 are shown to have characteristic positive-negative-positive residue stripes c.f. [38]. (**B**) Predicted α-helical regions of *Afr*LEA3m also showed characteristic positive-negative-positive residue stripes c.f. [38]. (**C**) *Af*LEA1 does not present the proposed charge pattern observed in *Afr*LEA2 and *Afr*LEA3m in its secondary structure. Still, its tertiary structure presents adjacent stripes of an alternating formal charge (positive-negative-positive-negative) on the protein surface.

**Figure 7 biomolecules-12-00425-f007:**
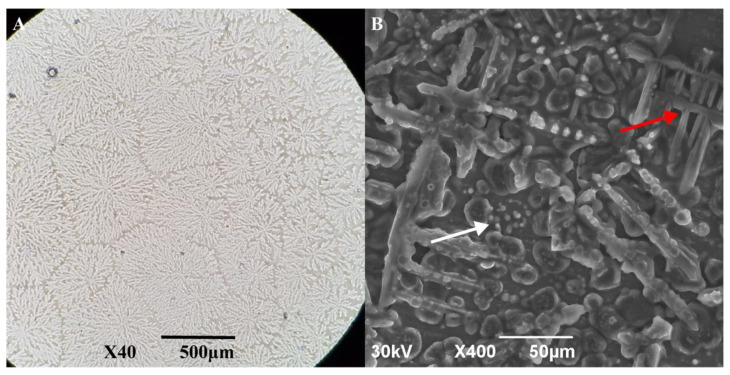
*Af*LEA1, dissolved in ultrapure water, crystallized readily when slowly desiccated at 80% RH. (**A**) Light microscopy shows that *Af*LEA1 dries into branching crystals. (**B**) Scanning electron microscopy (SEM) shows spherical structures (white arrow) are present in the desiccated sample in addition to the branching crystals (red arrow).

**Figure 8 biomolecules-12-00425-f008:**
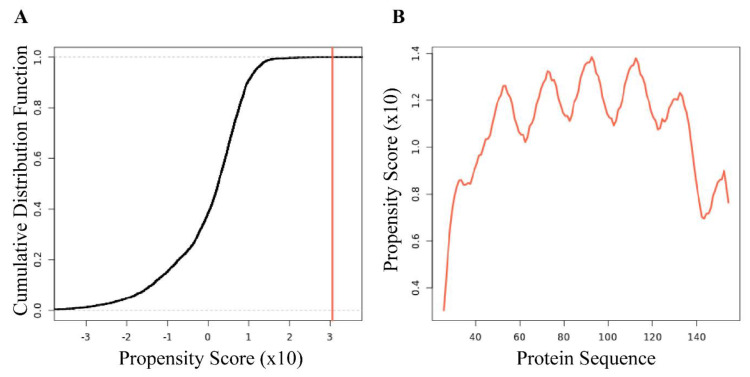
The bioinformatics program catGranule predicts that *Af*LEA1 has a high propensity towards liquid–liquid phase separation in the presence of RNA. (**A**) A cumulative distribution fraction analysis of the amino acids of *Af*LEA1 produces a propensity score of 3.05, and a score > 1 is a predictor of LLPS behavior. (**B**) The residue-level propensity of *Af*LEA1 to undergo LLPS, where values above 0 indicate an increased likelihood of undergoing LLPS in the presence of RNA.

**Figure 9 biomolecules-12-00425-f009:**
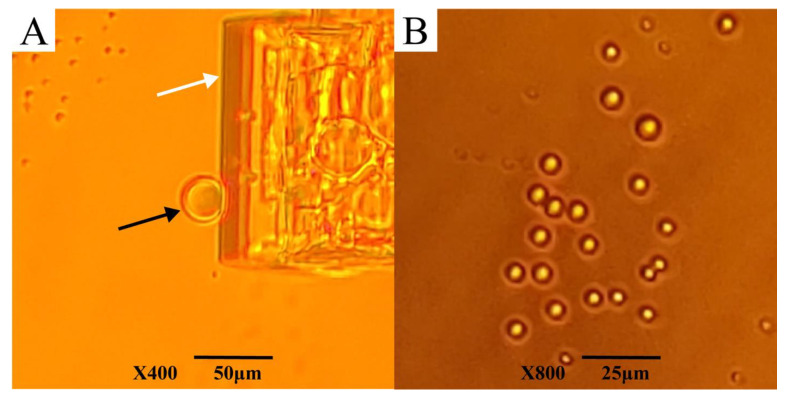
*Af*LEA1 undergoes LLPS during desiccation in vitro in a solution mimicking the intracellular conditions in *A. franciscana*. (**A**) When desiccated in the absence of RNA, *Af*LEA1 undergoes LLPS (black arrow) at severe dehydration concurrent with the formation of salt crystals in the solution (white arrow). (**B**) When desiccated in the presence of mRNA from *A. franciscana, Af*LEA1 rapidly undergoes LLPS at higher water contents than in an RNA-free solution and before salt crystals are formed.

**Figure 10 biomolecules-12-00425-f010:**
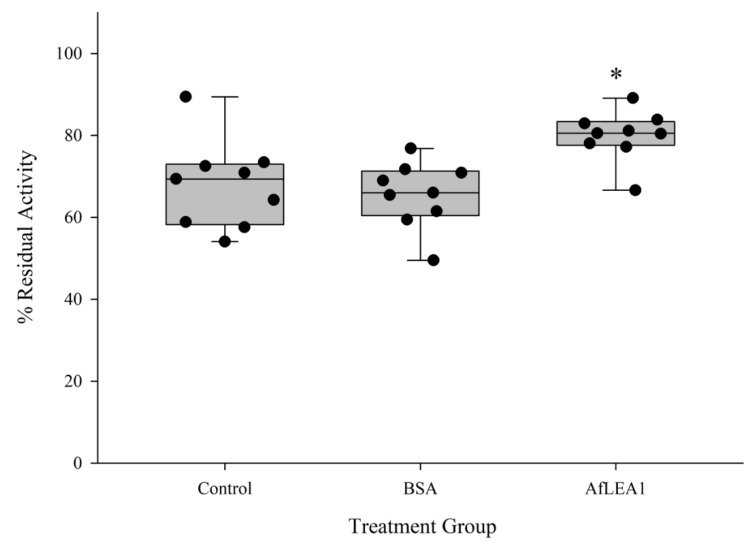
*Af*LEA1 (0.4 mg/mL) addition protects endogenous LDH activity in Kc167 cell lysates after desiccation and rehydration, while addition of BSA (0.4 mg/mL) did not significantly increases recovered LDH activity after rehydration compared to the control (*n* = 9; ±SD, *p* < 0.05; * different from control).

**Figure 11 biomolecules-12-00425-f011:**
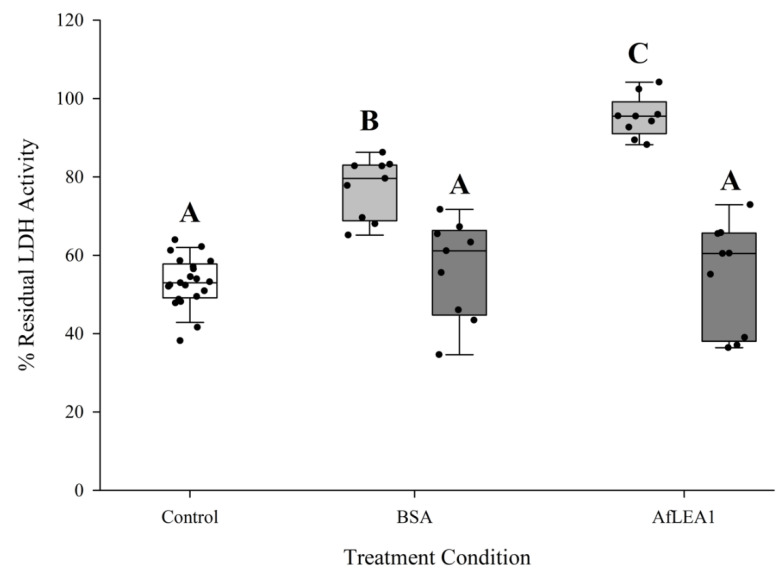
Residual activity of purified LDH was significantly increased when desiccated in the presence of BSA or *Af*LEA1 and then rehydrated with 100 mM sodium phosphate buffer at pH 6.5 (light gray box plots). LDH activity was not restored after desiccation in the absence of BSA or *Af*LEA1, and then rehydrated with 100 mM sodium phosphate buffer at pH 6.5 containing BSA or *Af*LEA1 (dark gray box plots). Control samples (white box plot) were desiccated and rehydrated in the absence of BSA or *Af*LEA1 (*n* = 9-18; ±SD, *p* < 0.05; different letters indicate significant differences between groups).

**Figure 12 biomolecules-12-00425-f012:**
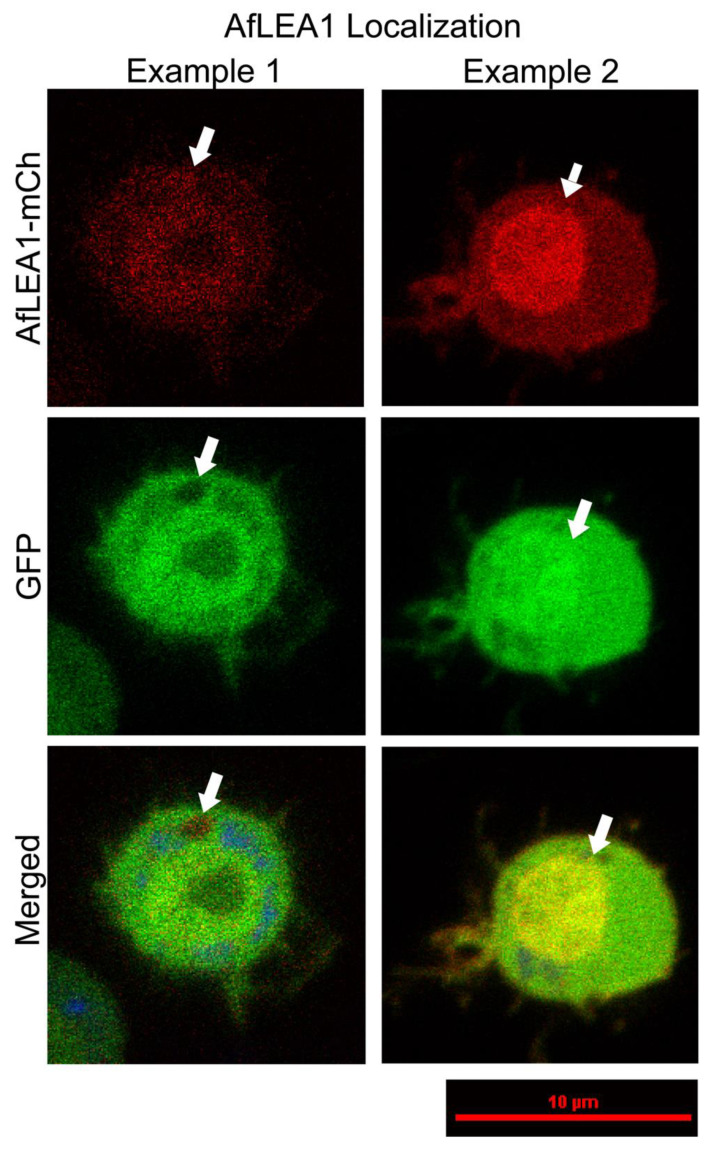
A chimeric protein composed of *Af*LEA1 and mCherry localizes predominately to the cytoplasm in *D. melanogaster* Kc167 cells ectopically co-expressing *Af*LEA1-mCherry and GFP (Examples 1 and 2). In some cells, *Af*LEA1-mCherry was also observed to undergo an LLPS and exclude GFP from the biomolecular condensate (Example 1). *Af*LEA1-mCherry was observed to accumulate in the nucleus of other cells (Example 2). Cells were stained with 200 µL of MitoView Blue, and *Af*LEA1-mCherry does not localize to the mitochondria. Images are representative images, and fluorescence intensities are not relative between examples (for a quantitative assessment, see Appendix A).

**Figure 13 biomolecules-12-00425-f013:**
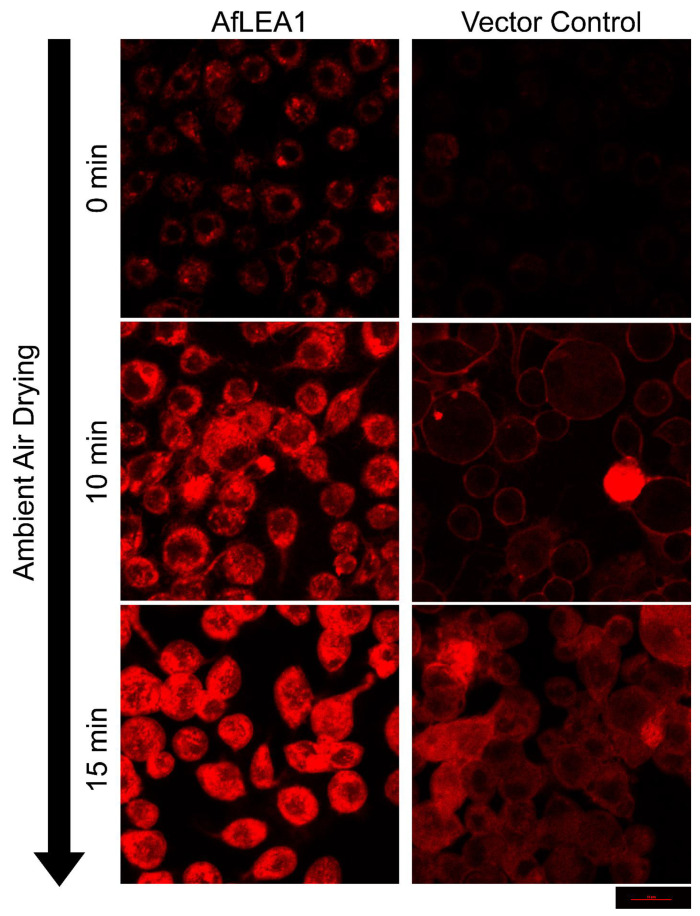
*Af*LEA1, when ectopically expressed in Kc167 cells, increases the intracellular viscosity, improves cellular integrity, and reduces plasma membrane fusions during desiccation. Cells ectopically expressing untagged *Af*LEA1, or the vector control, were stained for 2 min in 200 µL DPBS containing 0.1 µg/mL Nile Red (9-diethylamino-5H-benzo[a]phenoxazin-5-one). Cells were desiccated through evaporative water loss at ambient relative humidity. Fluorescence intensities are relative among all images.

## Data Availability

Data is contained within the article or Appendix A.

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
