# Peer review of "Functional and Conformational Plasticity of an Animal Group 1 LEA Protein"

_biomolecules, 2022, doi:10.3390/biom12030425_

Round 1

Reviewer 1 Report

The article focuses on proteins called Late Embryogenesis Abundant proteins (LEA). Group 3 LEA  proteins (LEA3s) are generally expressed in anhydrobiotic animals to guarantee tolerance to water-stress, while LEA proteins from group 1 (LEA1s) concurring to regulate anhydrobiosis only in Artemia franciscana. LEA1s differ from each other in the number of repetitions of an amino acid sequence called “motif of group 1” and therefore they show different lengths. Using different techniques, the authors investigate the secondary structure and the functional role in tolerance to water-deprivation of a specific and more abundant LEA1 of Artemia francescana, referred to as AfLEA1.

The article is well written, clear and adequately supported by scientific literature used as reference. The materials and methods are described in detail and the authors acted with procedural rigor. Furthermore, the results provide interesting insights and the variety of techniques used is appreciable. Nevertheless, some results obtained are promising but give rise to speculations which should be supported by more in-depth studies, as however correctly and promptly highlight by authors themselves.

My opinion is overall positive, but there are some points below that should be better clarified by the authors:

- The conformation plasticity of AfLEA1 was mainly investigated through bioinformatics techniques and secondary structure studies using circular dichroism. However, the authors have indications of a certain conformation plasticity of AfLEA1 already during its purification. In fact, the chromatography profile of anion exchange chromatography showed two peaks corresponding to the same protein (AfLEA1). The presence of the two peaks indicate that the protein binds the column with different affinities caused by a different charge density that, according to the authors, is due to two different conformations adopted by the protein. However, the theoretical pI of the protein (calculated using ProtParam and the sequence of AfLEA1 - ABR67402) is 7.93. The pI is therefore extremely close to the pH  of the buffer used (pH 8.0) in order to negatively charge the protein. The buffer must usually be at least 1 pH value higher than the pI of the protein in order to prevent non-specific interactions that could cause the presence of a double peak. In the light of that, can the authors explain how to interpret the double-peak?

- Furthermore, the authors note that there is a negative correlation between the amount of protein loaded in the column and the retention time - and, therefore, the concentration of NaCl used for elution. Observing Figure 2, the retention time changes exclusively because the elution starts before when more protein is loaded in the column. In particular, when more protein is loaded the resolution gets worse in terms of efficiency. Therefore, to confirm this negative correlation, the authors should exclude an overload of the column or a loading of more volume of the sample (please indicate whether the amount of sample loaded has been increased while maintaining the loaded volume unchanged).

-In Figure 2 please indicate the concentration of NaCl as a black line.

- Figure 3 I suggest to indicate the conalbumin and the carbonic anhydrase also with (2) and (3).

- One of the most interesting data of the article is the characterization of the secondary structure of AfLEA1 by circular dichroism under different conditions of water availability to investigate the conformational plasticity of the protein. Interestingly, in the hydrated state, the protein exhibits 40% of secondary structure - mainly alpha helix and beta sheet - while in the dried state it surprisingly shows a 100% of secondary structure, mainly alpha helix. Please indicate if there is a reference for the use of 2% SDS to validate its use as crowding agent suitable to confirm the functional role of this disorder to order transition.

- I suggest to add CD-spectra in the supplementary information - if possible.

- According to the authors, the prediction of the structure of AfLEA1 by I-Tasser allowed to confirm the secondary structure of dried AfLEA1 measured by circular dichroism, and to identify the spacial organisation of AfLEA1 able to stabilize by charge this high content of alpha helices. Since I was unable to identify this stabilizing charges motif (Pos-Neg-Pos-Neg) in the AfLEA1 structure, can the authors tell me if the repetition of the alternating charges in the space is present in the structural detail reported in 7C?

- In the legend of Figure 12, dark gray indicates the presence of BSA but according to me the dark gray should indicate the Residual Activity of DLH when BSA and LEA1 are added during rehydration process while the light grey when BSA and LEA1 are added during desiccation(/rehydratationt) process. If that is true, please modify the legend.

Author Response

We wish to thank the reviewer for the time invested in reviewing our work and the constructive criticism of our manuscript. All points raised have been addressed, and the manuscript has been significantly improved in the process. Please find a detailed response to all issues raised below.

Sincerely,

Drs. Michael A. Menze and Brett Janis

  1. The conformation plasticity of AfLEA1 was mainly investigated through bioinformatics techniques and secondary structure studies using circular dichroism. However, the authors have indications of a certain conformation plasticity of AfLEA1 already during its purification. In fact, the chromatography profile of anion exchange chromatography showed two peaks corresponding to the same protein (AfLEA1). The presence of the two peaks indicate that the protein binds the column with different affinities caused by a different charge density that, according to the authors, is due to two different conformations adopted by the protein. However, the theoretical pI of the protein (calculated using ProtParam and the sequence of AfLEA1 - ABR67402) is 7.93. The pI is therefore extremely close to the pH of the buffer used (pH 8.0) in order to negatively charge the protein. The buffer must usually be at least 1 pH value higher than the pI of the protein in order to prevent non-specific interactions that could cause the presence of a double peak. In the light of that, can the authors explain how to interpret the double-peak?

Thank you for drawing this to our attention and we apologize causing confusion involving the use of Alden Warner’s AfLEA protein (ABR67402, termed: “LEA5”) as reference point. The protein used in this study was the nearly identical protein previously referred to as Af1 in Marunde et al 2013. However, we felt that AfLEA1.1 was a bit cumbersome and refered to the protein here as AfLEA1. We have given the sequence below. Due to the nearly identical nature of the protein to the set of group 1 LEA proteins discovered by Dr. Warner’s lab it has not been uploaded to the Protein Database. However, the sequence has been published in Marunde et al, 2013 and Janis et al, 2018 (References 12 and 17, respectively, in the manuscript). The theoretical isoelectric point of this AfLEA1, according to EXPASY ProtParam, is 9.37. Therefore, we postulate that these two peaks indicate the presence of different conformations of AfLEA1. Interestingly, the peaks towards the end of the elution curve also contain mainly AfLEA1, but their quantities were low, and their purity was not as high as the first two peaks, so they were excluded from the following experiments.

>AfLEA1 (Previously Af1 to distinguish it from a mitochondrial-targeted protein variant)

MELSSSKLNRSIFKRRSKMSEQGKLSRQEAGQRGGQARAEQLGHAEGYVEMGRKGGQARAEQLGHEGYQEMGQKGGQARAEQLGTEGYQEMGQKGGQKRAEQLGHEGYQEIGQKGGQTRAEQLGTEGYQEMGQKGGQTRAEQLGHEGYVQMGKMGGEARKQQMSPEDYAAMGQKGGLARQK

  1. Furthermore, the authors note that there is a negative correlation between the amount of protein loaded in the column and the retention time - and, therefore, the concentration of NaCl used for elution. Observing Figure 2, the retention time changes exclusively because the elution starts before when more protein is loaded in the column. In particular, when more protein is loaded the resolution gets worse in terms of efficiency. Therefore, to confirm this negative correlation, the authors should exclude an overload of the column or a loading of more volume of the sample (please indicate whether the amount of sample loaded has been increased while maintaining the loaded volume unchanged).

We agree that the concentrations of protein are very close together and that, for the results to be robust enough to support our claims, a wider range of concentrations should be used. We have indicated that the volume of protein loaded onto the column was consistent regardless of the concentration of protein. These data have been reinterpreted as a notable trend that warrants further study and we have moved Fig. 2 into the supplemental files (now Supplemental Figure 1).

  1. In Figure 2 please indicate the concentration of NaCl as a black line.

In figure 2 (Now Supplemental Figure 1), the black line in the upper left corner is the NaCl concentration. This is clarified in the figure legend.

  1. Figure 3 I suggest to indicate the conalbumin and the carbonic anhydrase also with (2) and (3).

Thank you for bringing this to our attention. Conalbumin and carbonic anhydrase have been labeled with (2) and (3). Figure 3 is now Figure 2.

  1. One of the most interesting data of the article is the characterization of the secondary structure of AfLEA1 by circular dichroism under different conditions of water availability to investigate the conformational plasticity of the protein. Interestingly, in the hydrated state, the protein exhibits 40% of secondary structure - mainly alpha helix and beta sheet - while in the dried state it surprisingly shows a 100% of secondary structure, mainly alpha helix. Please indicate if there is a reference for the use of 2% SDS to validate its use as crowding agent suitable to confirm the functional role of this disorder to order transition.

We are also very excited by this data. Previous work using 2% SDS as a crowding agent on group 3 LEA proteins by Boswell et al, 2014 showed conformational transitions using circular dichroism. Some crowding agents, such as TFE, have also been used. However, we felt that these compounds can force proteins into α-helical conformations that are not always physiologically relevant. Therefore, we did not include a TFE condition.

  1. I suggest to add CD-spectra in the supplementary information - if possible.

A multiple spline plot of the CD-spectra has been added as supplemental Fig. 2.

  1. According to the authors, the prediction of the structure of AfLEA1 by I-Tasser allowed to confirm the secondary structure of dried AfLEA1 measured by circular dichroism, and to identify the spacial organisation of AfLEA1 able to stabilize by charge this high content of alpha helices. Since I was unable to identify this stabilizing charges motif (Pos-Neg-Pos-Neg) in the AfLEA1 structure, can the authors tell me if the repetition of the alternating charges in the space is present in the structural detail reported in 7C?

The Pos-Neg-Pos-Neg motif described in the text is represented as alternating charges along individual α-helices. Alternating charged residues in-phase on α-helices has been shown to stabilize helix formation, even in the absence of water. However, these helices align with one another in a tertiary arrangement that puts negative charges next to negative charges and positive charges next to positive charges. This results in a bundle of helices that, together, create a surface with alternating positive and negative residues. These are shown in Figure 7C as vertical, colored stripes that are perpendicular to the direction of the helices.

  1. In the legend of Figure 12, dark gray indicates the presence of BSA but according to me the dark gray should indicate the Residual Activity of DLH when BSA and LEA1 are added during rehydration process while the light grey when BSA and LEA1 are added during desiccation(/rehydratationt) process. If that is true, please modify the legend.

We apologize for our confusing graphical representation and have corrected this figure. The control group has been consolidated into a single white box plot and the legend has been thoroughly revised. Figure 11 is now Figure 10. It now reads as:

Figure 11: Residual activity of purified LDH was significantly increased when desiccated in the presence of BSA or AfLEA1 and then rehydrated with 100 mM sodium phosphate buffer at pH 6.5 (light gray box plots). LDH activity was not restored after desiccation in the absence of BSA or AfLEA1, then rehydrated with 100 mM sodium phosphate buffer at pH 6.5 containing BSA or AfLEA1 (dark gray box plots). Control samples (white box plot) were desiccated and rehydrated in absence of BSA or AfLEA1 (n = 3-4; ±SD, p<0.05; different letters indicate significant differences between groups).

Reviewer 2 Report

Review comment:

In this manuscript, Brett et.al reported and made a comprehensive characterization of AfLEA1 protein, involving the protein state, the secondary structure, protein structure prediction, LLPS behaviors and the cell localization. I would endorse this paper for publication, after the following points are addressed:

Major comment:

  1. In Figure 2 and Table S1. The author tries to indicate that” the concentration of salt required for elution of AfLEA1 is negatively correlated with the quantity of AfLEA1 bound to the column for both elution fractions.” But in Table S1, the note about the peak height and peak volume of peak 2 is wrong according to the figure2. There is almost no difference of all these peaks. Besides, the figure legend of figure 2 is “The elution maxima (retention time) of AfLEA1 is dependent on the amount of loaded protein.” If the author wants to clarify this, please use a relative wide range amount of protein to repeat this. In conclusion, current data cannot support what the author describe in the main text.

  1. In figure 3, the author further characterizes the two fractions from resource Q anion exchanger with SEC method suing Superdex200(10/300) column (method 2.3). To my knowledge, the Superdex200(10/300) should own quite different elution behavior for those markers. Please check is it Superdex200(10/300) or Superdex75(10/300). Otherwise, the data and information provide cannot support the results and analysis. Besides, it is will be a much better if the author can use these two fractions and run SEC-MALS for the absolute molar mass and size characterization. This will benefit the author to discuss about the conformational states.  

  1. In the result 3.3. the author’s analysis is based on the I-TASSER result. The I-TASSER will provide 10 reference structures hit from PDB. Which one the author used here. Did the author use the powerful Alphfold2 to do the prediction? Are they different from each other?

Optional major comments and suggestion:

  1. The author indicate AfLEA1 has an exceptionally high propensity to undergo LLPS, particularly when interacting with RNA. Would it be possible for the author verify the interaction of LEA1 and RNA or use a labeled RNA to verify it was incorporated into AfLEA1 dense phase?
  2. Result 3.4. “Unfortunately, scanning electron microscopy (SEM) reveals that the crystals formed by AfLEA1 during slow drying are of insufficient quality to employ X-ray crystallography to verify the structure of AfLEA1 in the desiccated state (Figure 8B).” Would it possible using the negative staining TEM to characterize this issue?

Minor comment:

  1. Page 3: “bacterial debris was removed by centrifugation for 30 min at 5,000 x g at 4ËšC.” 5,000xg should not be possible for bacterial debris separation.
  2. Result 1 “The presence of two elution peaks containing AfLEA1 protein that behaves identically during electrophoresis may indicate higher-order assemblies that are lost during the denaturation step in electrophoresis or distinct conformational states undetectable by SDS-PAGE.” The state of these two fractions is identical. As the author mentioned later, it also may be because of conformation difference.  The author may provide a native-page or cross-linking the protein at the same time.
  3. Result 3.2. the sub-title is ‘AfLEA1 Structural Analysis’. I prefer change it to ‘AfLEA1 Secondary Structural Analysis’.
  4. In figure 4, please indicate the meaning of different colors in the figure to make it clearer for readers.
  5. In figure 5, 6, 7. The positive charge of amino acid is usually colored with blue and red for negative ones in the field of structure biology. The author may revise the color labeling. The sticks model of the amino acid is hard to distinguish in the figures. The author may use Pymol software to generate a colorful one for better illustration.
  6. Result 3.3. “the I-Tasser predictions match within 2% the secondary structure content measured by circular dichroism for desiccated AfLEA1.” Only 2%? Compare with hydrated one or the dry one?
  7. Result 3.3. the last discussion paragraph is related with the major comment 2 above.
  8. Figure 12. The labeling and color are confusing. Please use different color and labels to make it much clearer.

Author Response

We wish to thank the reviewer for the time invested in reviewing our work and the constructive criticism of our manuscript. All points raised have been addressed, and the manuscript has been significantly improved in the process. Please find a detailed response to all issues raised below.

Sincerely,

Drs. Michael A. Menze and Brett Janis

Major comment:

  1. In Figure 2 and Table S1. The author tries to indicate that” the concentration of salt required for elution of AfLEA1 is negatively correlated with the quantity of AfLEA1 bound to the column for both elution fractions.” But in Table S1, the note about the peak height and peak volume of peak 2 is wrong according to the figure2. There is almost no difference of all these peaks. Besides, the figure legend of figure 2 is “The elution maxima (retention time) of AfLEA1 is dependent on the amount of loaded protein.” If the author wants to clarify this, please use a relative wide range amount of protein to repeat this. In conclusion, current data cannot support what the author describe in the main text.

We recognize that a larger range of protein concentrations should be used to create a more robust set of data necessary to support our claims and have deleted lines 372-382 from the manuscript. While we did identify a significant difference between the peak heights and elution volume, these concentrations are all quite close. We have moved Figure 2 to supplemental data (Supplemental Figure 1) and have reduced the impact of the figure on the text as a notable trend that, while interesting, requires additional research before we can draw meaningful conclusions from it.

2. In figure 3, the author further characterizes the two fractions from resource Q anion exchanger with SEC method suing Superdex200(10/300) column (method 2.3). To my knowledge, the Superdex200(10/300) should own quite different elution behavior for those markers. Please check is it Superdex200(10/300) or Superdex75(10/300). Otherwise, the data and information provide cannot support the results and analysis. Besides, it is will be a much better if the author can use these two fractions and run SEC-MALS for the absolute molar mass and size characterization. This will benefit the author to discuss about the conformational states.  

Thank you for catching this! You are correct – we used a Superdex 75(10/300). This has been changed in the methods section. We agree that we would love to run the protein using SEC-MALS to characterize the molecular weight and size. Unfortunately, this equipment is not available to us. Figure 3 is now Figure 2.

3. In the result 3.3. the author’s analysis is based on the I-TASSER result. The I-TASSER will provide 10 reference structures hit from PDB. Which one the author used here. Did the author use the powerful Alphfold2 to do the prediction? Are they different from each other?

The I-Tasser result shown was the first result, which also had the highest homogeny. However, each of the output files was virtually identical, the only difference being small changes to the orientations of the α-helices. Although Alphfoldl2 prediction is not currently available, we have compared it to similar PFAM_LEA5 proteins and found that Alphafold2 predicted a virtually identical α-helices for the PFAM_LEA5 motif. We chose to use I-Tasser to predict the desiccated state of AfLEA1 because it utilizes crystal structure data to do so. Most proteins are not natively folded in the anhydrous state, which introduces some level of inaccuracy to crystal structures. However, the structure of anhydrous AfLEA1 is precisely what we mean to predict. I-Tasser’s avoidance of confounding the structural prediction with force field dynamics associated with hydration layers was uniquely helpful in predicting the conformation of AfLEA1 in the dry state.

Optional major comments and suggestion:

  1. The author indicate AfLEA1 has an exceptionally high propensity to undergo LLPS, particularly when interacting with RNA. Would it be possible for the author verify the interaction of LEA1 and RNA or use a labeled RNA to verify it was incorporated into AfLEA1 dense phase?

At this time, we cannot conduct this experiment within the revision window of this submission. While we are confident that this phenomenon is not simply due to molecular crowding based on control groups containing large sucrose polymers, we stop short of claiming that the AfLEA1 dense phase contains RNA. This will be further examined in a follow-up study using both labeled RNA and super-charged green fluorescence proteins, as previously performed with AfrLEA6 (c.f., Belott et al., 2020). Still, the association between PFAM_LEA5 and nucleic acids has already been established in the literature. We feel that the association of phase separation with RNA is apparent with this simple experiment and that it is worth sharing our results with the scientific community.

  1. Result 3.4. “Unfortunately, scanning electron microscopy (SEM) reveals that the crystals formed by AfLEA1 during slow drying are of insufficient quality to employ X-ray crystallography to verify the structure of AfLEA1 in the desiccated state (Figure 8B).” Would it possible using the negative staining TEM to characterize this issue?

We believe that this should be possible. However, we do not currently have the technical expertise to perform this optimization. We have attempted to address this concern using panels of crystallization solutions, but we have yet to grow a crystal with high enough quality to perform X-ray crystallography.

Minor comment:

  1. Page 3: “bacterial debris was removed by centrifugation for 30 min at 5,000 x g at 4ËšC.” 5,000xg should not be possible for bacterial debris separation.

Thank you for catching this typo. This was the speed used to pellet bacterial cells during expression. Cellular debris were separated by centrifugation at 20,000 x g at 4 °C.

  1. Result 1 “The presence of two elution peaks containing AfLEA1 protein that behaves identically during electrophoresis may indicate higher-order assemblies that are lost during the denaturation step in electrophoresis or distinct conformational states undetectable by SDS-PAGE.” The state of these two fractions is identical. As the author mentioned later, it also may be because of conformation difference.  The author may provide a native-page or cross-linking the protein at the same time.

Thank you for helping us elaborate better. The SDS PAGE was used to demonstrate that there was not a population of breakdown products, nor was there any indication of oligomerization. Folding-induced oligomerization has been hypothesized for some LEA proteins. We feel that size-exclusion chromatography is sufficient to address the absence of higher-order oligomers.

  1. Result 3.2. the sub-title is ‘AfLEA1 Structural Analysis’. I prefer change it to ‘AfLEA1 Secondary Structural Analysis’.

We agree that this is a more accurate sub-title. Result 3.2 has been renamed to ‘AfLEA1 Secondary Structure Analysis.’

  1. In figure 4, please indicate the meaning of different colors in the figure to make it clearer for readers.

We have added a guide to the colors in Figure 4 (Now Figure 3) at the beginning of the legend. It now reads:

      1. Circular dichroism analysis of hydrated and desiccated AfLEA1, wherein α-helices (red), β-sheets (green), turns (blue), and random coils (magenta) are represented as proportions of the protein. In the hydrated state, the secondary structure of AfLEA1 was on average 5% α-helices, 35% β-sheets, 18% turns, and 42% random coils. In the desiccated state, the secondary structure of AfLEA1 was on average 85% α-helices, 5% β-sheets, 10% turns, and 0% random coils. In the presence of 2% SDS, the secondary structure of AfLEA1.1 was on average of 25% α-helices, 13% β-sheets, 16% turns, and 46% random coils.
  1. In figure 5, 6, 7. The positive charge of amino acid is usually colored with blue and red for negative ones in the field of structure biology. The author may revise the color labeling. The sticks model of the amino acid is hard to distinguish in the figures. The author may use Pymol software to generate a colorful one for better illustration.

Thank you for drawing attention to this mistake on our part. We have changed the colors to reflect the conventions of the field of structural biology in figures 5,6 and 7 (now 4, 5, and 6, respectively). We investigated using Pymol to generate a better illustration, but we do not currently have a license for the software. To help make the colors more distinct, the background color has been changed to black.

  1. Result 3.3. “the I-Tasser predictions match within 2% the secondary structure content measured by circular dichroism for desiccated AfLEA1.” Only 2%? Compare with hydrated one or the dry one?

We agree that the wording of this section can use improvement. The point that we wished to convey is that the I-Tasser predictions for each secondary structure of AfLEA1 was within 2% of the measured value using circular dichroism in the desiccated state. This sentence now reads as follows:

    1. The program is surprisingly effective at predicting the ordered arrangements of LEA proteins that undergo conformational transitions during desiccation, and the secondary structure predictions of I-Tasser all fall within 2% the secondary structure content measured by circular dichroism for desiccated AfLEA1.
  1. Result 3.3. the last discussion paragraph is related with the major comment 2 above.

We have addressed the concern regarding the Superdex 75(10/300) column. This is corrected in the methods section. However, we will not be able to conduct SEC-MALS analysis at this time.

8. Figure 12. The labeling and color are confusing. Please use different color and labels to make it much clearer.

After reviewing this legend, we believe that we missed a change in the figure during final drafting. We believe that consolidating the control group into a single box and correcting the figure legend has addressed the confusion of the figure. Figure 12 is now figure 11. It now reads:

Figure 11: Residual activity of purified LDH was significantly increased when desiccated in the presence of BSA or AfLEA1 and then rehydrated with 100 mM sodium phosphate buffer at pH 6.5 (light gray box plots). LDH activity was not restored after desiccation in the absence of BSA or AfLEA1, then rehydrated with 100 mM sodium phosphate buffer at pH 6.5 containing BSA or AfLEA1 (dark gray box plots). Control samples (white box plot) were desiccated and rehydrated in absence of BSA or AfLEA1 (n = 3-4; ±SD, p<0.05; different letters indicate significant differences between groups).

Round 2

Reviewer 2 Report

The author made effort to answer the revision comments. There are still some points need to be further addressed and please do not try to ignore these comments. I would endorse this paper for publication only after the following points are addressed:

  1. As I mentioned in my previous Major comment 1, the Table S1 is wrong. The Elution volume and peak height of Elution peak2 is not correct. Please check and revise it.
  2. In the previous minor comment 2, I mentioned the native-page or cross-linking the protein or some other method is good choice for strong the conclusion of the state of AfLEA1 since the limitation of SEC method. This is an easy work and I strongly suggest the author do it.
  3. In the answer of the minor comments 5, The license is not an excuse for the deficiency of the work. The pymol can be registered as an academic user or buy the license. and I strongly suggest the author do it. The structure figures is really not profession as shown in the current version.

Author Response

Response to the reviewer:

  1. As I mentioned in my previous Major comment 1, Table S1 is wrong. The Elution volume and peak height of Elution peak2 is not correct. Please check and revise it.

We thank the reviewer for bringing this to our attention. We have corrected Supplemental Table 1.

  1. In the previous minor comment 2, I mentioned the native-page or cross-linking the protein or some other method is good choice for strong the conclusion of the state of AfLEA1 since the limitation of SEC method. This is an easy work and I strongly suggest the author do it. 

We have performed some initial experiments using native PAGE to gain further insights into the conformational assembly of AfLEA1. However, the results were not immediately insightful. We agree that additional studies should be performed in the future and are planning to optimize blue native PAGE to investigate the behavior of multiple group 1 LEA proteins with different numbers of group LEA 1 motifs.

  1. In the answer of the minor comments 5, The license is not an excuse for the deficiency of the work. The pymol can be registered as an academic user or buy the license. and I strongly suggest the author do it. The structure figures is really not profession as shown in the current version.

We respectfully disagree with the reviewer. However, we have purchased a license for pymol and restructured some of the graphs. However, potentially due to us not being skilled in using the program, the output files were not improved over the files obtained using PDB viewer.